# Identification of a novel non-invasive biological marker to overcome the shortcomings of PSA in diagnosis and risk stratification for prostate cancer: Initial prospective study of developmental endothelial locus-1 protein

**Jae-Wook Chung[1]◉, Hyun Tae Kim[1]◉, Yun-Sok Ha[1], Eun Hye Lee[2], So Young Chun[2], Chan-Hyeong Lee[3], Kyeong Hyeon Byeon[1], Seock Hwan Choi[1], Jun Nyung Lee[1], Bum Soo Kim[1], Tae-Hwan Kim[1], Eun Sang Yoo[1], Ghil Suk Yoon[4], Moon-Chang Baek[3‡]\*, Tae Gyun Kwon[1,5‡]\***

**1** Department of Urology, School of Medicine, Kyungpook National University, Daegu, Republic of Korea, **2** Biomedical Research Institute, Kyungpook National University, Daegu, Republic of Korea, **3** Department of Molecular Medicine, Cell and Matrix Research Institute, School of Medicine, Kyungpook National University, Daegu, Korea, **4** Department of Pathology, School of Medicine, Kyungpook National University, Daegu, Republic of Korea, **5** Joint Institute for Regenerative Medicine, Kyungpook National University, Daegu, Republic of Korea

◉ These authors contributed equally to this work.
‡ MCB and TGK also contributed equally to this work.
\* mcbaek@knu.ac.kr (MCB); tgkwon@knu.ac.kr (TGK)

## Abstract

### Objective

This prospective study sought to clarify the developmental endothelial locus-1 (Del-1) protein as values of diagnosis and risk stratification of prostate cancer (PCa).

### Design

From February 2017 to December 2019, a total 458 patients who underwent transrectal ultrasound guided prostate biopsy or surgery of benign prostatic hyperplasia agreed to research of Del-1 protein. We prospectively compared and analyzed the Del-1 protein and prostate specific antigen (PSA) in relation to the patients' demographic and clinicopathological characteristics.

### Results

Mean age was 68.86±8.55 years. Mean PSA and Del-1 protein was 21.72±89.37, 0.099 ±0.145, respectively. Two hundred seventy-six (60.3%) patients were diagnosed as PCa. Among them, 181 patients underwent radical prostatectomy (RP). There were significant differences in Del-1 protein between benign and PCa group (0.066±0.131 vs 0.121±0.149, respectively, p<0.001). When we set the cut-off value of del-1 protein as 0.120, in patients

**Data Availability Statement:** All relevant data are within the manuscript and its Supporting Information files.

**Funding:** This work was supported by Biomedical Research Institute grant, Kyungpook National University Hospital (2019).

**Competing interests:** The authors have declared that no competing interests exist.

**Abbreviations:** PCa, prostate cancer; PSA, prostate-specific antigen; Del-1, developmental endothelial locus-1; BPH, benign prostatic hyperplasia; TURP, transurethral resection of prostate; HoLEP, holmium laser enucleation of prostate; TRUS, transrectal ultrasound; RP, radical prostatectomy; ADT, androgen deprivation therapy; ELISA, enzyme-linked immunosorbent assay; HR, hazard ratio; CI, confidence internal; EV, extracellular vesicle; IHC, immunohistochemistry; ROC, receiver operating characteristic.

with $3 \leq PSA \leq 8$, positive predictive value and specificity of Del-1 protein ($\geq 0.120$) for predicting PCa were 88.9% (56/63) and 93.5% (101/108), respectively. Among 181 patients who underwent RP, there were significant differences in Del-1 protein according to stage (pT2 vs pT3a vs $\geq$pT3b) (0.113±0.078, 0.171±0.121, 0.227±0.161, respectively, p<0.001) and to Gleason score (6 (3+3) or 7 (3+4) vs 7 (4+3) or 8 (4+4) vs 9 or 10) (0.134±0.103, 0.150±0.109, 0.212±0.178, respectively, $P = 0.044$). Multivariate analysis showed that PSA, Del-1 protein and high Gleason score ($\geq 9$) were the independent prognostic factors for predicting higher pT stage ($\geq$3b). Furthermore, age, PSA and Del-1 protein were independent prognostic factors for predicting significant PCa.

## Conclusion

Patients with PCa showed higher expression of Del-1 protein than benign patients. Del-1 protein increased with the stage and Gleason score of PCa. Collaboration with PSA, Del-1 protein can be a non-invasive useful marker for diagnosis and risk stratification of PCa.

## Introduction

Serum prostate-specific antigen (PSA) is the most widely used biomarker for the early detection and management of PCa. The diagnosis of prostate cancer (PCa) is precipitated by a persistent increase in the serum PSA level, which triggers the performance of a prostate biopsy. However, PSA screening is controversial due to its relative lack of cancer specificity. Approximately 20% to 30% of patients with potential PCa were not diagnosed as having PCa at the time of the initial prostate biopsy [1]. Furthermore, some study has shown that only 32.4% of patients with a PSA 4.0 ng/mL or higher had PCa [2]. Thus, current PSA screening is not cost-effective from a public health perspective [3] and many urologists are concerning for over-diagnosis of PCa with widespread serum PSA screening, which eventually could lead to overtreatment of PCa and its accompanied morbidities. And also, prostate biopsy under transrectal ultrasound (TRUS) is the only tool for diagnosing PCa, but it is invasive and can cause side effects such as infection and gross hematuria, and can give considerable discomfort to the patients during procedure.

To overcome these drawbacks of PSA, numerous biological markers have being developed to enhance the specificity of PCa screening such as free PSA, prostate health index, 4K score, prostate cancer antigen 3, SelectMDx®, ConfirmMDx®, *TMPRSS2:ERG* GENE FUSION, and ExoDx PROSTATE INTELLISCORE [4, 5]. However, these markers are expensive and rarely refunded in the general care system, thus these are rarely used in clinical practice. Given the limitations of PSA, the invasiveness of prostate biopsy and over-treatment of low-risk PCa and when considering that the need for excavating of new useful biomarkers is emerging [6], we focused on serum developmental endothelial locus-1 (Del-1) protein.

In 2016, our center already published the Del-1 protein which is located in circulating extracellular vesicles as a novel biomarker for early breast cancer detection [7]. Considering the similarity of breast cancer and PCa, we hypothesized that serum Del-1 protein may be related to PCa and investigated whether Del-1 protein has any association with PCa.

## Materials and methods

### Ethics statement

This study was approved by the institutional review board of Kyungpook National University, School of Medicine, Daegu, republic of Korea (IRB Number KNUH 2017-02-017). The study

was carried out in agreement with the applicable laws and regulations, good clinical practices, and ethical principles as described in the Declaration of Helsinki. All patients gave their written informed consent after a thorough explanation of the study procedure involved.

## Rationale

To achieve the scientific rationality of this study which focused on elevated serum Del-1 protein in patients with PCa, we performed the following process before the start of this study; 1) Identification of differences in Del-1 protein expression between benign prostatic tissue and PCa tissue, 2) Confirmation of decrease of Del-1 protein expression after radical prostatectomy (RP), and 3) Previous study which demonstrated the elevated Del-1 protein level in breast cancer that shared similar properties to PCa [7].

## Study design

Fig 1 shows study algorithm. From February 2017 to December 2019, 610 patient agreed to participate in this prospective study. Of these patients, 152 patients who withdrew informed consent during the study or did not visit regularly were excluded and finally, 458 patients were included in present study. 19 patients were diagnosed as benign after surgery of benign prostatic hyperplasia (BPH) such as transurethral resection of prostate (TURP) or holmium laser enucleation of prostate (HoLEP). 439 patients underwent TRUS guided prostate biopsy due to elevated PSA ($\geq$3.0 ng/mL). Of these patients, 182 patients were diagnoses as no tumor and 276 patients as prostate adenocarcinoma. 181 patients underwent radical prostatectomy (RP) and 95 patients received androgen deprivation therapy (ADT) or radiation therapy or docetaxel based chemotherapy.

Furthermore, we subdivided the patients who underwent RP into 'clinically insignificant PCa' and 'significant PCa' group according to RP specimen. Clinically insignificant PCa group must meet all three criteria below and may be eligible for active surveillance [8, 9]; 1) PSA<10 ng/dL, 2) Gleason 6 (3+3) or 7 (3+4), 3) pT stage 2. There were 42 patients who satisfied these three criteria. The remaining 139 patients were classified into significant PCa group.

Present study is the first fundamental step for biomarker development which is the identification of high-quality samples to perform high-quantity analysis, including proteomics and expression array analysis [10, 11]. In this study, a large amount of PCa patients' serum was used for discovery of new biomarker before meta-analysis.

## Method for detection of serum Del-1 protein

Fig 2 shows enzyme-linked immunosorbent assay (ELISA) method for detecting Serum Del-1 protein [7]. For quantification of serum Del-1 protein on extracellular vesicle (EV), 96-well

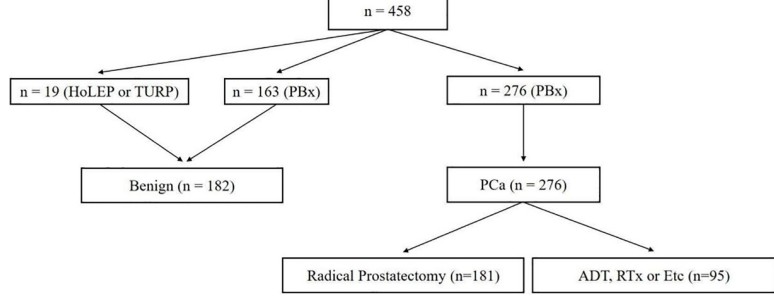

**Fig 1. Study design.** HoLEP, holmium laser enucleation of prostate; TURP, transurethral resection of prostate; PBx, prostate biopsy; PCa, prostate cancer; ADT, androgen deprivation therapy; RTx, radiation therapy.

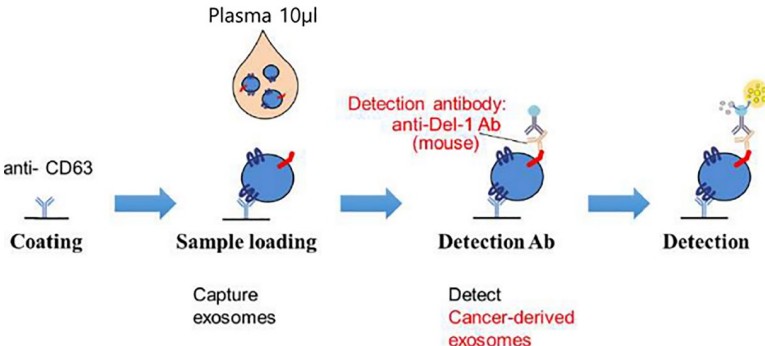

**Fig 2. Enzyme-linked immunosorbent assay (ELISA) for detecting developmental endothelial locus-1 (Del-1) protein.**

plates were coated overnight with polyclonal anti-CD63 (ab68418; Abcam) antibody at 200 ng/well in 0.2 mol/L sodium phosphate buffer (pH 6.5). The plates were blocked for 2 hours at room temperature with 200 μL of phosphate buffered saline (PBS) containing 1% bovine serum albumin (BSA) and washed three times with PBS containing 0.05% Tween 20 (PBS-T). Serum was diluted 1:10 with blocking buffer. The diluted serum (100 μL) was added to the plates in triplicate and incubated for 2 hours at room temperature. And then the plates were reacted for 2 hours with monoclonal anti–Del-1 (ab88667; Abcam) antibody at 200 ng/well and washed five times with PBS-T. The plates were reacted for 1 hour with peroxidase-conjugated anti-mouse IgG antibody. After washing with PBS-T seven times, the plates were developed with 3,3′,5,5′-tetramethylbenzidine containing hydrogen peroxide. The reaction was stopped with 1 mol/L phosphoric acid, and optical density values were measured at 450 nm on an automated iMark plate reader (BioRad).

## Method for detection of Del-1 protein expression in prostatic tissue

The prostate tissue samples were fixed in 10% formalin solution to form paraffin tissue blocks through dehydration, clearing and embedding processes. The tissue blocks were sectioned into 4μm thickness. Paraffin was removed by clearing process and hydrated in distilled water, and the antigen was retrieved in pH 6.0 citrate buffer solution before permeabilization with Triton-x solution. Followed by permeabilization, tissue slides were applied with blocking solution before being stained with anti-del-1 antibody (Abcam, Cambridge, MA, USA) and andi-CD63 (Novus Biologicals, Littleton, CO, USA) for overnight at 4°C. FITC labelled secondary antibodies were applied for 2 h at room temperature and the slides were mounted by using mounting solution with DAPI (Vector Laboratories, Burlingame, CA, USA).

## Statistical analysis

Continuous variables (age, PSA, Del-1 protein) were analyzed by a Student's *t* test or analysis of variance (ANOVA). Comparisons of non-continuous variables (categorical Del-1 protein, Gleason score, pT stage) were analyzed by chi-square test or Fisher's exact test. Cut-off value for Del-1 protein was determined using receiver operating characteristic (ROC) curve analysis. Multivariate logistic regression analysis was used for predicting advanced disease (pT $\geq$ 3b) and significant PCa. Statistical analysis was performed using SPSS 16.0 for Windows (SPSS Inc., Chicago, IL, USA), and a p value of $<0.05$ was considered statistically significant.

**Table 1.  Basic characteristics of the patients (benign versus prostate cancer).**

|  | Total | Benign | PCa | *p* value |
|---|---|---|---|---|
| N | 458 (100.0) | 182 (39.7) | 276 (60.3) | - |
| Age | 68.86±8.55 | 64.49±9.12 | 71.08±7.36 | <0.001 |
| Body mass index | 24.26±2.62 | 24.30±2.61 | 24.23±2.63 | 0.771 |
| Prostate volume | 35.78±15.24 | 34.96±13.83 | 36.32±16.11 | 0.336 |
| PSA | 21.72±89.37 | 6.11±7.83 | 32.64±113.86 | <0.001 |
| PSA density | 0.80±3.91 | 0.21±0.41 | 1.18±4.99 | 0.001 |
| Del-1 protein | 0.099±0.145 | 0.066±0.131 | 0.121±0.149 | <0.001 |
| Del-1 protein |  |  |  | <0.001 |
| <0.120 | 331 | 168 | 163 |  |
| ≥0.120 | 127 | 14 | 113 |  |
| Non-regional lymph node or distant metastasis | - | - | 32 (11.59) | - |

## Results

Table 1 and Fig 3 show basic characteristics of the patients with benign and PCa. Mean age was 68.86±8.55years and there were significant differences in patients' age between the two groups (64.49±9.12 vs 71.08±7.36, p<0.001). Mean body mass index was 24.26±2.62 kg/m$^2$ and prostate volume measured by TRUS was 35.78±15.24 mL. Mean PSA was 21.72±89.37 ng/mL and there were significant differences in PSA between the two groups (6.11±7.83 vs 32.64 ±113.86, p<0.001). Mean PSA density was 0.80±3.91ng/mL$^2$ and there were significant differences between the two groups (0.21±0.41 vs 1.18±4.99, p = 0.001). Mean Del-1 protein was 0.099±0.145 ng/mL and there were significant differences in PSA between the two groups (0.066±0.131 vs 0.121±0.149, p<0.001). Among the PCa patients, 32 patients (11.59%) showed non-regional lymph node or distant metastasis.

The best cut-off value of Del-1 protein was computed to be 0.120 in accordance with the ROC curve. And the area under the ROC curve was 0.680 (95% CI 0.632–0.728; p<0.001)

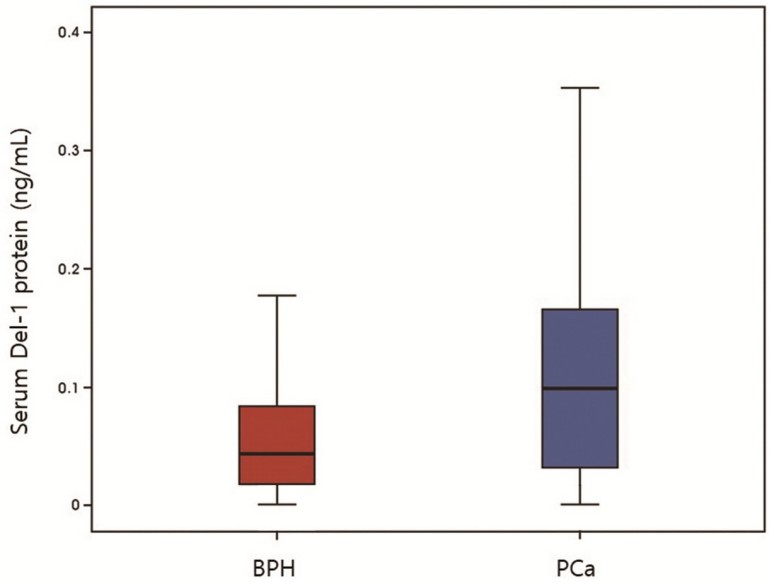

**Fig 3. Basic characteristics of the patients (benign versus prostate cancer).** PCa, prostate cancer.

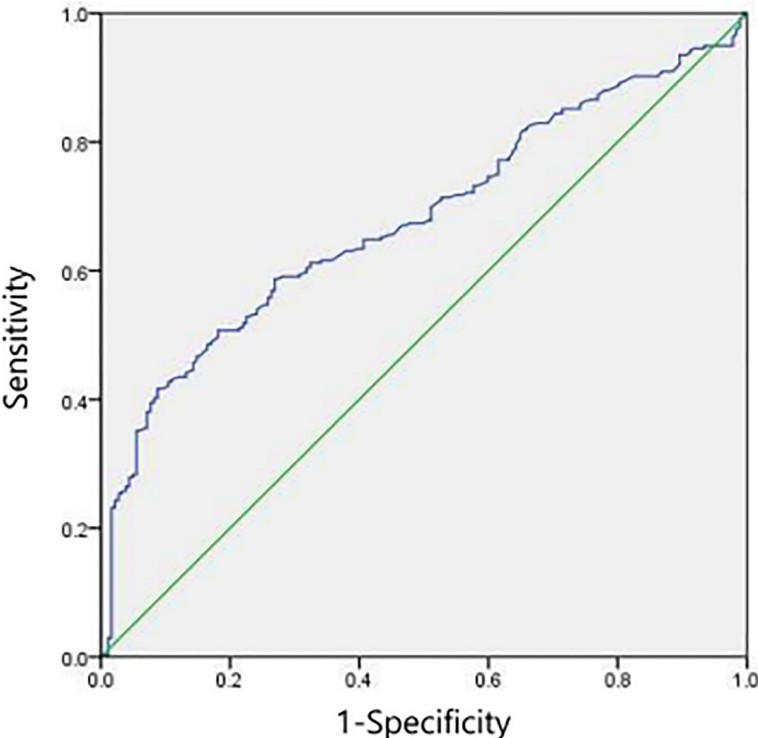

**Fig 4. The best cut-off value for detecting developmental endothelial locus-1 (Del-1) protein according to the receiver operating characteristic curve.**

(Sensitivity: 41.7%, Specificity: 90.7%) (Fig 4). PCa group had higher portion of patients with Del-1 protein≥0.120 than benign group (p<0.001). Among the 182 patients diagnosed as benign, only 14 (7.7%) patients showed Del-1 protein 0.120 or higher.

Fig 5 shows immunohistochemistry (IHC) results in benign prostate tissue and prostate adenocarcinoma according to pT stage and Gleason score. In IHC analysis using anti-Del-1 antibody (or anti-CD63 antibody), expression of Del-1 protein was higher in PCa tissue than in benign tissue. Furthermore, expression of Del-1 protein increased as the cancer stage and Gleason score increased.

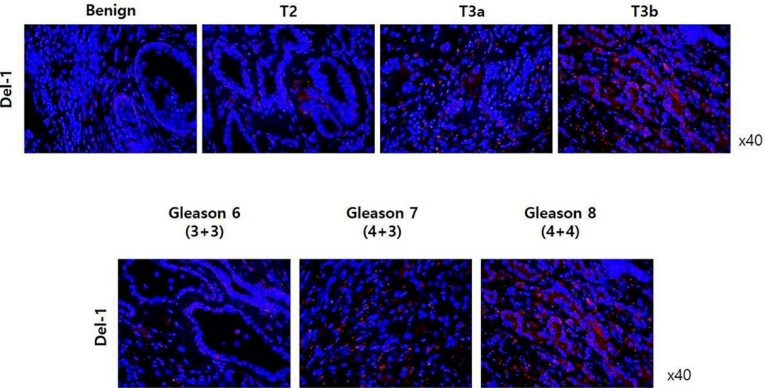

**Fig 5. Immunohistochemistry results in in benign prostate tissue and prostate adenocarcinoma according to pT stage and Gleason score.**

**Table 2. Distribution of patients when cut-off value of Del-1 protein and PSA is 0.120 and 3.0.**

|  | Total | Benign | PCa | *p* value |
|---|---|---|---|---|
| PSA≥3 | 406 | 142 | 264 | - |
| Del-1 protein |  |  |  | <0.001 |
| Del-1<0.120 | 284 | 132 | 152 |  |
| Del-1≥0.120 | 122 | 10 | 112 |  |
| PSA<3 | 52 | 40 | 12 |  |
| Del-1 protein |  |  |  | 0.999* |
|  | 47 | 36 | 11 |  |
| Del-1≥0.120 | 5 | 4 | 1 |  |

* Fisher's exact test

Table 2 is the distribution of patients when cut-off value of Del-1 protein and PSA is set at 0.120 and 3.0 ng/mL, respectively. When PSA is 3 or more, PCa group had a significant high proportion of patient with Del-1≧0.120 (p<0.001) and positive predictive value of Del-1≧0.120 for predicting PCa is 91.8% (112/122). When PSA is less than 3, specificity of Del-1<0.120 was 90.0% (36/40).

Table 3 shows the distribution of patients when PSA is set between 3 and 8 (PSA gray zone). PCa group had a significant high proportion of patient with Del-1≧0.120 (p<0.001) and positive predictive value of Del-1≧0.120 for predicting PCa is 88.9% (56/63). Specificity of Del-1<0.120 was 93.5% (101/108).

Fig 6A and 6B shows level of Del-1 protein according to pT stage (A) and Gleason score (B) in 181 patients who underwent RP. 100 patients were diagnosed as pT2, 56 as pT3a, 25 as ≥ pT3b. PSA level increased significantly with the higher pT stage (6.47±3.75, 11.71±8.90, 16.10±16.67, respectively, p<0.001) (Table 4A). Furthermore, there were also significant differences in Del-1 protein according to stage (0.113±0.078, 0.171±0.121, 0.227±0.161, respectively, p<0.001) (Table 4A). 95 patients were diagnosed as Gleason 6 (3+3) or 7 (3+4), 71 as 7 (4+3) or 8 (4+4), 15 as 9 or 10. PSA level increased significantly with the higher Gleason score (6.70±3.94, 11.86±11.83, 15.14±11.41, respectively, p<0.001) (Table 4B). Furthermore, there were also significant differences in Del-1 protein according to Gleason score (0.134±0.103, 0.150±0.109, 0.212±0.178, respectively, p = 0.044) (Table 4B).

Table 5 shows multivariate logistic regression analysis of De1-1 protein as prognostic marker for predicting advanced disease (pT ≥ 3b). The multivariate analysis has shown that PSA (hazard ratio [HR], 1.045; 95% confidence internal [CI], 1.000–1.092; p = 0.050), Del-1 protein (HR, 87.833; 95% CI, 2.453–5060.403; p = 0.015), Gleason score (6 (3+3) or 7 (3+4) vs 9 or 10) (HR, 6.273; 95% CI, 1.516–25.419; p = 0.011) were independent prognostic factors for predicting advanced disease (pT ≥ 3b).

Table 6 shows multivariate logistic regression analysis of De1-1 protein as prognostic marker for predicting significant PCa. Age (HR, 1.045; 95% confidence internal [CI], 1.012–

**Table 3. Distribution of patients within PSA gray zone (3≤PSA≤8).**

|  | Total | Benign | PCa | *p* value |
|---|---|---|---|---|
| 3≤PSA≤8 | 237 | 108 | 129 | - |
| Del-1 |  |  |  | <0.001 |
| Del-1<0.120 | 174 | 101 | 73 |  |
| Del-1≥0.120 | 63 | 7 | 56 |  |

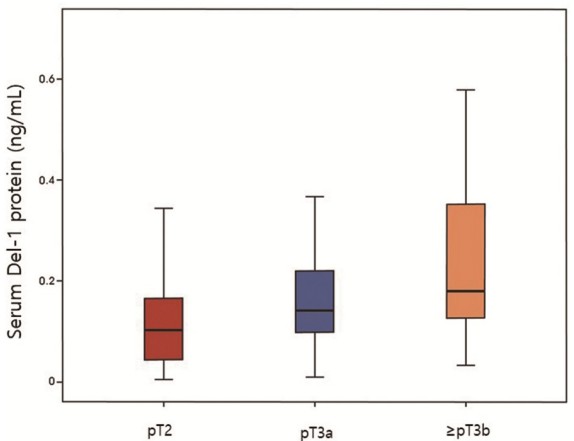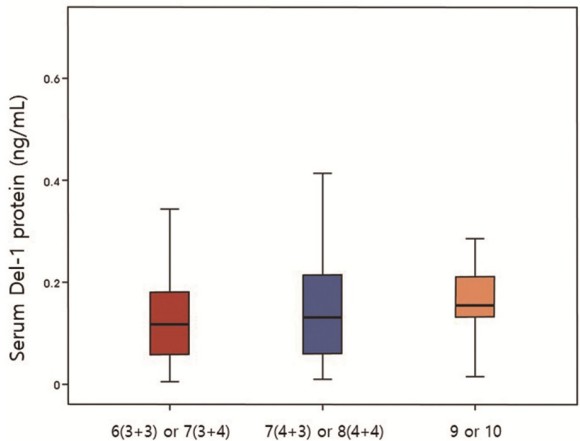

**Fig 6.** Developmental endothelial locus-1 protein according to (A) pT stage and (B) Gleason score in 181 patients who underwent radical prostatectomy.

1.079; p = 0.007), PSA (HR, 1.277; 95% CI, 1.136–1.435; p<0.001), Del-1 protein (HR, 22.935; 95% CI, 2.248–234.001; p = 0.008) were independent prognostic factors for predicting significant PCa.

Fig 7 shows the change of PSA and Del-1 protein in 14 patients who underwent RP and did not show biochemical recurrence. There were significant decrease between preoperative and postoperative 12 months serum PSA (9.80±9.32 vs 0.02±0.02, p = 0.002) and Del-1 protein (0.062±0.039 vs 0.029±0.025, p = 0.015) (Table 7).

## Discussion

Since PSA's first introduction in the late 1980s, it has dramatically changed the natural course of PCa from a cancer which urologists used to detect somewhat later that we were difficult to offer appropriate curable treatments to a cancer which urologists detect too early that we are uncertain about how to manage it [6]. As PSA in being used widely around the world, unnecessary prostate biopsies are increasing which has eventually lead to an overtreatment of low risk PCa resulting in several unwarranted compromises to urinary and sexual quality of life such as incontinence and erectile dysfunction. Currently, especially in developed countries,

**Table 4. Developmental endothelial locus-1 protein according to (A) pT stage and (B) Gleason score in 181 patients who underwent radical prostateectomy.**

| (A) pT stage | | | | |
|---|---|---|---|---|
| | **pT2** | **pT3a** | **≥ pT3b** | **p value** |
| N | 100 | 56 | 25 | - |
| Age | 69.76±7.44 | 71.73±6.43 | 72.48±6.09 | 0.999 |
| PSA | 6.47±3.75 | 11.71±8.90 | 16.10±16.67 | <0.001 |
| Del-1 protein | 0.113±0.078 | 0.171±0.121 | 0.227±0.161 | <0.001 |
| (B) Gleason score | | | | |
| | **6 (3+3) or 7 (3+4)** | **7 (4+3) or 8 (4+4)** | **9 or 10** | **p value** |
| N | 95 | 71 | 15 | - |
| Age | 70.87±6.96 | 70.30±7.30 | 72.07±6.30 | 0.655 |
| PSA | 6.70±3.94 | 11.86±11.83 | 15.14±11.41 | <0.001 |
| Del-1 protein | 0.134±0.103 | 0.150±0.109 | 0.212±0.178 | 0.044 |

**Table 5. Multivariate logistic regression analysis of De1-1 protein as prognostic marker for advanced disease (pT ≥ 3b).**

| | Total | pT2 or pT3a | ≥ pT3b | *p* value | | HR (95% CI) |
|---|---|---|---|---|---|---|
| | | | | Univariate | Multivariate | |
| N | 181 | 156 | 25 | - | - | - |
| Age | 70.75±7.02 | 70.47±7.14 | 72.48±6.09 | 0.184 | - | - |
| PSA | 9.42±9.03 | 8.35±6.59 | 16.10±16.67 | 0.030 | 0.050 | 1.045 (1.000–1.092) |
| Del-1 protein | 0.147±0.114 | 0.134±0.099 | 0.227±0.161 | 0.009 | 0.015 | 87.833 (2.453–5060.403) |
| Gleason score | | | | <0.001 | | |
| 6 (3+3) or 7 (3+4) | 95 | 88 | 7 | | | Ref |
| 7 (4+3) or 8 (4+4) | 71 | 60 | 11 | | 0.373 | 1.639 (0.553–4.854) |
| 9 or 10 | 15 | 8 | 7 | | 0.011 | 6.273 (1.516–25.419) |

PCa is characterized by its high incidence but overall low mortality. Therefore, to compensate for the weakness of PSA, novel diagnostic and prognostic markers are urgently needed.

Del-1 protein, also called epidermal growth factor (EGF)-like repeats and discoidin I-like domain-3 (EDIL-3), is a 52 kDa extracellular matrix associated glycoprotein released from endothelial cells which was identified and isolated from the embryonic mouse lung in 1998 [12]. Del-1 protein is known to be have an anti-inflammatory effect. In 2008, Choi et al. revealed that Del-1 protein is an endogenous inhibitor of inflammatory cell recruitment and it could provide a basis for targeting leukocyte-endothelial interactions in several disease [13]. Since 2008, numerous studies were conducted to investigate the function of Del-1 such as homeostatic factor in the central nervous system by limiting neuro-inflammation and demyelination [14], inhibitor of ischemia related angiogenesis by blocking inflammation [15], inhibitory factor for preventing peritoneal adhesion [16] and biomarker of endothelial dysfunction, sepsis, and sepsis induced organ dysfunction [17].

Del-1 is known to be highly expressed in the brain and lung, however it is less expressed in the kidney, liver, intestine, spleen and blood in mice [13]. Furthermore, several studies have shown that Del-1 protein is associated with tumor angiogenesis and plays an important role in interaction between cancer cells and endothelial cells [18–20]. Considering that chronic inflammation is a crucial mechanism for potentially leading to tumor development and progression, inhibition of inflammation could be important therapeutic strategy for cancer treatment [21]. Recent study demonstrated that Del-1 protein is associated with various cancers including bladder cancer, colorectal cancer, hepatocellular carcinoma, and pancreatic cancer [22–25]. However, the precise mechanism between the Del-1 protein and cancer is still in debate.

Extracellular vesicle (EV) is membrane vesicles released by numerous cells. Tumor cells and neighboring cells in the cancer microenvironment, release EVs which have functions of

**Table 6. Multivariate logistic regression analysis of De1-1 protein as prognostic marker for predicting significant prostate cancer.**

| | Total (n = 363) | Benign (n = 182) + Insignificant PCa (n = 42) | Significant PCa (n = 139) | *p* value | | HR (95% CI) |
|---|---|---|---|---|---|---|
| | | | | Univariate | Multivariate | |
| Age | 68.66±8.77 | 66.77±9.11 | 71.70±7.23 | <0.001 | 0.007 | 1.045 (1.012–1.079) |
| PSA | 18.88±91.43 | 6.15±7.11 | 39.40±145.46 | 0.001 | <0.001 | 1.277 (1.136–1.435) |
| Body mass index | 24.23±2.62 | 24.26±2.54 | 24.19±2.75 | 0.812 | - | - |
| Prostate volume | 35.35±15.09 | 35.64±14.81 | 34.90±15.57 | 0.655 | - | - |
| PSA Density | 0.70±4.00 | 0.21±0.37 | 1.48±6.38 | 0.003 | 0.010 | 0.026 (0.002–0.409) |
| Del-1 protein | 0.103±0.130 | 0.080±0.130 | 0.140±0.123 | <0.001 | 0.008 | 22.935 (2.248–234.001) |

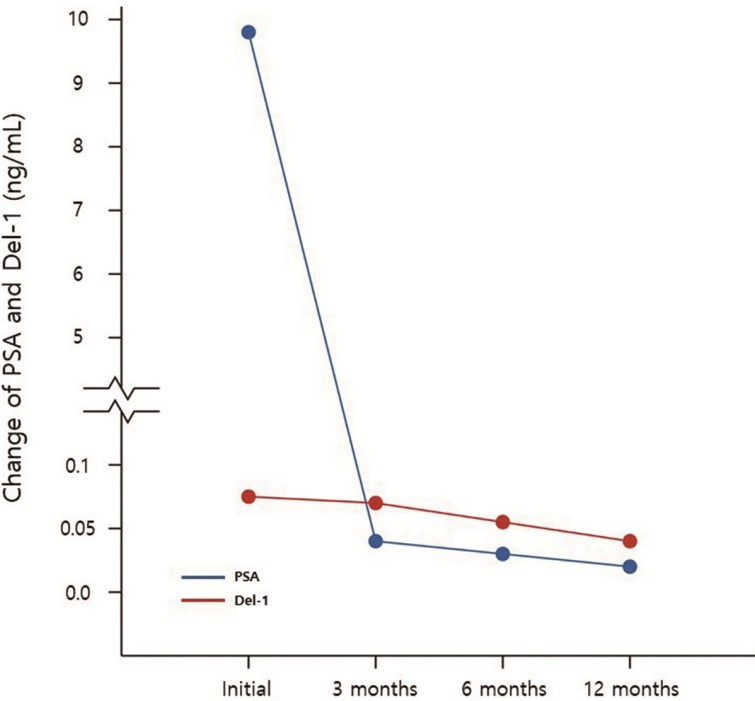

**Fig 7. Change of PSA and Del-1 protein in 14 patients who underwent radical prostatectomy and did not show biochemical recurrence.**

angiogenesis, pro-metastasis signaling, and suppressing immune system in an autocrine or paracrine manner [26–28]. Tumor cell–derived EVs may have the potential useful values to be used a biological markers for early detection of various cancers because these membranous vesicles are released continuously into blood from early stages of cancer [29, 30]. Furthermore, a few studies exhibited the function of the Del-1 protein in tumor cells [24]. Del-1 protein is expressed in some cancer cells [18, 24, 31] proposing that it might have potential roles as tumor-specific biological marker for numerous cancers, including PCa. Furthermore, in 2014, Liu et al. evaluated that PCa-derived exosomes are highly embedded with cancer marker molecules, especially membranous proteins including prostate specific membrane antigen, advocating the characteristics of the original PCa cells. Therefore, identification of cancer derived exosomes can provide the crucial evidences for the research of novel cancer related biological markers [32].

On the bases of aforementioned various studies, we hypothesized that EV proteins, especially Del-1 protein, can be used as diagnostic biological markers for PCa. We applied a proteomic attempt to identify Del-1 protein as tumor specific proteins on circulating EV isolated from the patients' serum with benign prostatic tissue as well as various stages of PCa. In 2016,

**Table 7. Change of PSA and Del-1 protein in 14 patients who underwent radical prostatectomy and did not show biochemical recurrence.**

|  | Initial | POD 3 months | POD 6 months | POD 12 months | *p* value |
|---|---|---|---|---|---|
| PSA | 9.80±9.32 | 0.04±0.05 | 0.03±0.02 | 0.02±0.02 | 0.002* |
| Del-1 protein | 0.062±0.039 | 0.058±0.047 | 0.049±0.062 | 0.029±0.025 | 0.015** |

* Comparison of initial and POD 12 month PSA

** Comparison of initial and POD 12 month Del-1 protein

our team evaluated the association between Del-1 protein and breast cancer [7]. Del-1 protein level in serum was calculated by ELISA from healthy controls (n = 81), breast cancer patients (n = 269), breast cancer patients after surgical resection (n = 50), patients with benign breast tumors (n = 64), and patients with noncancerous diseases (n = 98) in two cohorts. Expression of Del-1 protein was significantly higher in breast cancer patients than in all controls (p<0.001) and normalized after surgical resection. The diagnostic accuracy of Del-1 protein measured by area under the ROC curve, 0.961 [95% CI, 0.924–0.983], sensitivity of 94.70%, and specificity of 86.36% in test cohort and 0.968 (0.933–0.988), 92.31%, and 86.62% in validation cohort for early-stage breast cancer. Furthermore, Del-1 protein maintained diagnostic accuracy for patients with early-stage breast cancer using the other type of ELISA [0.946 (0.905–0.972), 90.90%, and 77.14% in the test cohort; 0.943 (0.900–0.971), 89.23%, and 80.99% in the validation cohort].

PCa and breast cancer are both solid malignancies of accessory sex organs and have characteristics of being epithelial primarily, hormone-driven cancers, which respond well to endocrine treatment [33]. PCa and breast cancer are both affected significantly by steroid hormones and removal or suppressing of gonads can reduces the risk of cancer development dramatically in both sexes. Anti-estrogens are beneficial and possibly preventive for breast cancer while anti-androgens are beneficial and possibly preventive for PCa. Considering these biological similarity that breast cancer and PCa both are hormone-sensitive disease [34] and are related to androgen receptor pathway [35], we presumed that Del-1 protein would be highly expressed in patients with PCa than benign prostatic tissue. First, to confirm the expression of Del-1 level in prostatic tissue, we performed the IHC stain in patients with BPH and PCa. IHC revealed that Del-1 level of PCa group was significantly higher than benign group. Furthermore, in IHC stain, expression of Del-1 increased as the pT stage increased. Based on this experimental evidence derived from prostate tissue level, we also presumed that serum Del-1 protein would be highly expressed in patients with PCa than benign or health control group. As the same result, serum Del-1 protein in patients with PCa was higher than benign or health control group and expression of serum Del-1 protein was significantly increased with higher pT stage.

To our best knowledge, present trial is the first study to demonstrate the diagnostic and risk stratification values of Del-1 protein on serum EV as a novel biomarker for PCa. Combination of measurement of serum Del-1 protein with serum PSA may enhance the identification of patients with PCa and may reduce unnecessary prostate needle biopsy.

However, present study has some limitation. There are no evidences of Del-1's superiority over PSA, and Del-1 could be valuable only when collaboration with PSA. Decline of Del-1 expression not only after PR but also radiation therapy or medical treatment such as ADT or docetaxel based chemotherapy should be evaluated. Furthermore, we cannot guarantee that patients of benign group is 'real' benign. Because there can be hidden cancer in patients of benign group whose first prostate biopsy is negative or whose biopsy results are 'no tumor' after TURP or HoLEP. Another problem is that detecting Del-1 protein takes a lot of time and cost in current technology. Too small in number of healthy patients also should be considered and therefore the conclusions should be interpreted with caution. The biggest practical difficulties when considering which biological marker for PCa to use is the absence of prospective head to head trials comparing the various biomarkers from numerous trials [6]. Furthermore, previous some trials have requested that even within a given test the cutoff value used in one population may not be the most appropriate cutoff value for another population [36, 37]. Therefore, in the future, further large-scale population-based multi-institutional prospective trials for PCa biomarkers including Del-1 protein should be performed.

## Conclusions

There are imminent academic expectation of identifying novel biomarkers with the capability to more precisely diagnose PCa due to lack of cancer specificity of PSA. Present study is the first trial to demonstrate the diagnostic and risk stratification value of Del-1 protein on serum EV as a non-invasive novel marker for PCa. Patients with PCa showed higher expression of Del-1 protein than benign patients. Del-1 protein increased with the stage and Gleason score of PCa. Collaboration with PSA, Del-1 protein can be a useful marker for diagnosis and risk stratification of PCa.

## Supporting information

**S1 File.**
(ZIP)

## Author Contributions

**Conceptualization:** Jae-Wook Chung, Hyun Tae Kim, Moon-Chang Baek, Tae Gyun Kwon.

**Data curation:** Yun-Sok Ha, Eun Hye Lee, So Young Chun.

**Formal analysis:** Chan-Hyeong Lee, Kyeong Hyeon Byeon, Seock Hwan Choi, Ghil Suk Yoon.

**Investigation:** Jun Nyung Lee, Bum Soo Kim, Tae-Hwan Kim, Eun Sang Yoo.

**Methodology:** Jae-Wook Chung, Bum Soo Kim, Tae-Hwan Kim, Eun Sang Yoo.

**Resources:** Jae-Wook Chung, Eun Hye Lee, So Young Chun, Jun Nyung Lee.

**Supervision:** Hyun Tae Kim, Moon-Chang Baek, Tae Gyun Kwon.

**Validation:** Jae-Wook Chung.

**Visualization:** Hyun Tae Kim.

**Writing – original draft:** Jae-Wook Chung.

**Writing – review & editing:** Hyun Tae Kim, Moon-Chang Baek, Tae Gyun Kwon.

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
