## [Decision Letter · Decision Letter 0]

19 Feb 2021

PONE-D-21-02189

Identification of a novel non-invasive biological marker to overcome the shortcomings of PSA in diagnosis and risk stratification for prostate cancer: Initial prospective study of developmental endothelial locus-1 protein

PLOS ONE

Dear Dr. Kwon,

Thank you for submitting your manuscript to PLOS ONE. After careful consideration, we feel that it has merit but does not fully meet PLOS ONE’s publication criteria as it currently stands. Therefore, we invite you to submit a revised version of the manuscript that addresses the points raised during the review process.

We look forward to receiving your revised manuscript.

Kind regards,

Jeremy Yuen Chun Teoh

Academic Editor

PLOS ONE

Journal Requirements:

2.We suggest you thoroughly copyedit your manuscript for language usage, spelling, and grammar. If you do not know anyone who can help you do this, you may wish to consider employing a professional scientific editing service.  

3.Thank you for stating the following in the Acknowledgments Section of your manuscript:

"This research was supported by the Basic Science Research Program through the National Research Foundation of Korea (NRF) & funded by the Korean government (MSIT) (2018R1C1B5040264) (2019R1A2C1004046) (2019R1F1A1044473) (2019R1H1A1079839)."

4.PLOS requires an ORCID iD for the corresponding author in Editorial Manager on papers submitted after December 6th, 2016. Please ensure that you have an ORCID iD and that it is validated in Editorial Manager. To do this, go to ‘Update my Information’ (in the upper left-hand corner of the main menu), and click on the Fetch/Validate link next to the ORCID field. This will take you to the ORCID site and allow you to create a new iD or authenticate a pre-existing iD in Editorial Manager. Please see the following video for instructions on linking an ORCID iD to your Editorial Manager account: https://www.youtube.com/watch?v=_xcclfuvtxQ

<h1>** **</h1>

Additional Editor Comments:

Overall a nice paper investigating the value of Del-1 protein concentration in prostate cancer.

1. Provide more information in the patient characteristics, e.g. DRE finding, any metastasis

2. Usually we classify prostate cancer into clinically significant vs insignificant cancers. Additional analysis especially multivariate analysis is needed for this outcome.

3. For the Del-1 protein cut-off value, how does its diagnostic performance compare with PSA?

4. Does Del-1 protein have any implications on the oncological outcomes?

Reviewers' comments:

Reviewer's Responses to Questions

**Comments to the Author**

1. Is the manuscript technically sound, and do the data support the conclusions?

Reviewer #1: Yes

2. Has the statistical analysis been performed appropriately and rigorously? 

Reviewer #1: I Don't Know

3. Have the authors made all data underlying the findings in their manuscript fully available?

Reviewer #1: Yes

4. Is the manuscript presented in an intelligible fashion and written in standard English?

Reviewer #1: Yes

5. Review Comments to the Author

Reviewer #1: This is great work looking into early stages of developing a new biomarker in diagnosis of PCa.

I believe the manuscript can be written in a better way. Introduction and discussion is lengthy. I suggest in the introduction focus on limitation of PSA briefly and introduce the new biomarker. This should be one or two paragraph. The discussion also can be more to the point. Great work mentioning the limitation. In addition, indicate that this work is in what stage of biomarker development- there are know phases of developing biomarkers.

6. PLOS authors have the option to publish the peer review history of their article (what does this mean?). If published, this will include your full peer review and any attached files.

Reviewer #1: No

---

## [Author Response · Author response to Decision Letter 0]

25 Feb 2021

[Jeremy Yuen Chun Teoh, Academic Editor]

★ After careful consideration, we feel that it has merit but does not fully meet PLOS ONE’s publication criteria as it currently stands. Therefore, we invite you to submit a revised version of the manuscript that addresses the points raised during the review process.

Please submit your revised manuscript by Apr 05 2021 11:59PM.

★ � I updated the statement of my financial disclosure in my cover letter.

★ Guidelines for resubmitting your figure files are available below the reviewer comments at the end of this letter.

★ If applicable, we recommend that you deposit your laboratory protocols in protocols.io to enhance the reproducibility of your results. Protocols.io assigns your protocol its own identifier (DOI) so that it can be cited independently in the future. For instructions see: http://journals.plos.org/plosone/s/submission-guidelines#loc-laboratory-protocols

It is inappropriate to deposit my laboratory protocols and data because there are a lot of data related to personal informations.

We ask for your understanding. 

[Journal Requirements]

I’ve changed my manuscript according to PLOS ONE's style.

I send the files that have been corrected by a professional proofreader in English (Enago) as a ‘supporting information’ file.

"This research was supported by the Basic Science Research Program through the National Research Foundation of Korea (NRF) & funded by the Korean government (MSIT) (2018R1C1B5040264) (2019R1A2C1004046) (2019R1F1A1044473) (2019R1H1A1079839)."

We note that you have provided funding information that is not currently declared in your Funding Statement. However, funding information should not appear in the Acknowledgments section or other areas of your manuscript. We will only publish funding information present in the Funding Statement section of the online submission form. Please remove any funding-related text from the manuscript and let us know how you would like to update your Funding Statement. Currently, your Funding Statement reads as follows:

I removed the funding-related text from the manuscript. Instead, we put the Funding Statement in my cover letter.

[This work was supported by Biomedical Research Institute grant, Kyungpook National University Hospital (2019).]

I’ve updated my ORCID iD.

[Additional Editor Comments]

Overall a nice paper investigating the value of Del-1 protein concentration in prostate cancer.

1. Provide more information in the patient characteristics, e.g. DRE finding, any metastasis

Thank you for valuable comment. 

I’ve added additional information about the patients’ characteristics, including BMI, prostate volume, PSA density and presence of non-regional lymph node or distant metastasis.

However, a lot of data about DRE finding were missing. 

We ask for your understanding.

2. Usually we classify prostate cancer into clinically significant vs insignificant cancers. Additional analysis especially multivariate analysis is needed for this outcome.

Thank you for valuable comment. 

1) https://pubmed.ncbi.nlm.nih.gov/20653654/

2) https://pubmed.ncbi.nlm.nih.gov/31715271/

We subdivided the patients who underwent radical prostatectomy into clinically insignificant PCa and significant PCa group according to RP specimen.

Clinically insignificant PCa group must meet all three criteria below and may be eligible for active surveilance; 1) PSA<10 ng/dL, 2) Gleason 6 (3+3) or 7 (3+4), 3) pT stage 2. 

And then, we finally performed multivariate analysis among the patients with benign plus insignificant PCa versus the patients with significant PCa.

3. For the Del-1 protein cut-off value, how does its diagnostic performance compare with PSA?

Thank you for valuable comment. 

As we already described in Results (Table 2, 3) & Discussion section, cut-off value of Del-1 protein as 0.120 showed high specificity and positive predictive value. However, in this regard alone, we cannot guarantee that Del-1 protein is superior to PSA. 

Only when collaboration with PSA, Del-1 protein can be useful in terms of reducing unnecessary PBx.

4. Does Del-1 protein have any implications on the oncological outcomes?

Thank you for valuable comment. 

We strongly believed that Del-1 protein is associated with BCR.

And we already found that Del-1 protein decreased after RP.

However, as the number of patients who were investigated for the relationship between BCR and Del-1 protein is too small, we may be able to provide the exact facts in the next paper.

We ask for your understanding.

[Reviewers' comments to the Author]

Reviewer #1: This is great work looking into early stages of developing a new biomarker in diagnosis of PCa.

I believe the manuscript can be written in a better way. Introduction and discussion is lengthy. I suggest in the introduction focus on limitation of PSA briefly and introduce the new biomarker. This should be one or two paragraph. The discussion also can be more to the point.

Thank you for valuable comment. 

I have made short and stylish changes to the introduction and conclusion section.

Great work mentioning the limitation. In addition, indicate that this work is in what stage of biomarker development-there are known phases of developing biomarkers.

Thank you for valuable comment. 

https://www.ncbi.nlm.nih.gov/pmc/articles/PMC1584291/pdf/neo0801_0059.pdf

Present study is the first fundamental step for biomarker development which is the identification of high-quality samples to perform high-quantity analysis, including proteomics and expression array analysis. In this study, a large amount of PCa patients’ serum was used for discovery of new biomarker before meta-analysis.

I added this point to the manuscript.

---

## [Decision Letter · Decision Letter 1]

5 Apr 2021

Identification of a novel non-invasive biological marker to overcome the shortcomings of PSA in diagnosis and risk stratification for prostate cancer: Initial prospective study of developmental endothelial locus-1 protein

PONE-D-21-02189R1

Dear Dr. Kwon,

We’re pleased to inform you that your manuscript has been judged scientifically suitable for publication and will be formally accepted for publication once it meets all outstanding technical requirements.

Kind regards,

Jeremy Yuen Chun Teoh

Academic Editor

PLOS ONE

---

## [Editor Report · Acceptance letter]

15 Apr 2021

PONE-D-21-02189R1 

Identification of a novel non-invasive biological marker to overcome the shortcomings of PSA in diagnosis and risk stratification for prostate cancer: Initial prospective study of developmental endothelial locus-1 protein 

Dear Dr. Kwon:

I'm pleased to inform you that your manuscript has been deemed suitable for publication in PLOS ONE. Congratulations! Your manuscript is now with our production department. 

Kind regards, 

on behalf of

Dr. Jeremy Yuen Chun Teoh 

Academic Editor

PLOS ONE